# CD163 Antibodies Inhibit PRRSV Infection via Receptor Blocking and Transcription Suppression

**DOI:** 10.3390/vaccines8040592

**Published:** 2020-10-09

**Authors:** Huiling Xu, Zehui Liu, Suya Zheng, Guangwei Han, Fang He

**Affiliations:** 1Institute of Preventive Veterinary Medicine, College of Animal Sciences, Zhejiang University, Hangzhou 310058, China; 11917012@zju.edu.cn (H.X.); zehuiliu@zju.edu.cn (Z.L.); 21917039@zju.edu.cn (S.Z.); 11617032@zju.edu.cn (G.H.); 2Zhejiang Provincial Key Laboratory of Preventive Veterinary Medicine, Hangzhou 310058, China

**Keywords:** PRRSV, CD163, monoclonal antibody, antiviral strategy

## Abstract

CD163 has been identified as the essential receptor for Porcine reproductive and respiratory syndrome (PRRSV), a major etiologic agent of pigs. Scavenger receptor cysteine-rich domain 5–9 (SRCR5–9) in CD163 was shown to be responsible for the virus interaction. In this study, monoclonal antibodies (mAbs) 6E8 and 9A10 against SRCR5–9 were selected based on the significant activity to inhibit PRRSV infection in Porcine Alveolar Macrophage (PAMs) and Marc-145. Both mAbs are capable of blocking variable PRRSV strains in a dose-dependent manner. Meanwhile, as candidates for both prevention and therapeutics, the antibodies successfully inhibit PRRSV infection and the related NF-κB pathway either before or after virus attachment. Besides, the antibody treatment with either mAb leads to a remarkable decrease of CD163 transcription in PAMs and Marc-145. It is potentially caused by the excessive accumulation of membrane associated CD163 due to the failure in CD163 cleavage with the antibody binding. Further, conformational epitopes targeted by 6E8 and 9A10 are identified to be spanning residues ^570^SXDVGXV^576^ in SRCR5 and Q^797^ in SRCR7, respectively. CD163 with mutated epitopes expressed in 3D4 cells fails to support PRRSV infection while wild type CD163 recovers PRRSV infection, indicating the critical role of these residues in PRRSV invasion. These findings promote the understanding in the interaction between PRRSV and the receptor and provide novel broad antiviral strategies for PRRSV prevention and treatment via alternative mechanisms.

## 1. Introduction

Porcine reproductive and respiratory syndrome (PRRS) is a highly contagious viral disease that has a significant economic impact on the pig industry worldwide [1]. The causative agent, PRRS virus (PRRSV), is a single-stranded, positive-sense RNA virus, belonging to the family Arteriviridae with the order Nidovirales [2,3]. Multiple PRRSV lineages coexist in swine herds including lineages 1, 3, 5, and 8 in China [4]. The genome of PRRSV is approximately 15.4 kb in length which encodes six structural proteins, including the envelope protein E, membrane protein M, and glycoproteins (GPs) GP2 (or GP2a), GP3, GP4, and GP5, and a replication and transcription complex [5]. PRRSV replication triggers the activation of NF-κB, ERK1/2, and p38MAPKs, which results in the release of proinflammatory cytokines in host against virus infection [6,7]. However, PRRSV is characterized by high variation, persistent infection, delayed neutralizing antibody response, suppressed NK cell, and antibody-dependent enhancement to counter host immunity. To the date, PRRS has not been effectively controlled [8,9,10,11,12]. Both modified live and inactivated PRRS vaccines failed to provide sustainable disease control especially when faced with a heterologous PRRSV strain [13]. Therefore, effective multiple heterologous PRRSV control and prevention strategies are urgently needed.

PRRSV has a restricted host and cell tropism, with porcine alveolar macrophages (PAMs) as the major target [14,15]. PRRSV infect host cells relying on specific cellular receptors and endocytosis to complete the viral life cycle. The interactions between viral structural proteins and cellular receptors are thought to determine the tissue tropism and host range for viruses. To date, four independent but functionally related PRRSV receptors have been reported. Sialoadhesin (Sn) have been identified to mediate internalization for PRRSV, which interacts with M/GP5 complex [16,17]. Heparan sulfate (HS) serves as an attachment factor for PRRSV [18,19]. Most importantly, scavenger receptor CD163 is the key factor for PRRSV infection via promoting viral uncoating and internalization in macrophages [20,21]. Recently, experiments in vivo with CD163 gene edited pigs (CD163-null pigs, CD163 domain swapping pigs, and CD163 SRCR5 deletion pigs) confirmed that CD163 is essential for PRRSV infection as the major receptor [22,23].

CD163 is a 130 kDa type-I transmembrane glycoprotein that is exclusively expressed on the surface of PAMs and green monkey kidney cell lines (MA-104 and MARC-145) [14,15,24]. Its extracellular domain is composed of nine scavenger receptor cysteine-rich (SRCR) domains of about 100–110 amino acid residues [25]. CD163 is the main factor regulating the internal homeostatic balance, mainly adjusting of cell-free hemoglobin (Hb) and Hb/haptoglobin complexes [26,27]. Binding of the hb-hp complex to CD163 molecule induces the production of cytokines such as IL-10, which in turn stimulates the production of CD163 and heme enzyme [28]. Inversely, CD163 expression is down-regulated by proinflammatory cytokines including IL-1α, IL-1β, IL-4, IL-8, tumor necrosis factor-alpha (TNF-α), and even lipopolysaccharide (LPS) [12]. Most recently, it is proposed that GP4 can recruit GP2a, GP3, and GP5 to form a complex and promote the interaction of GP2a/GP4 binding CD163 [29]. Further, deleted and chimeric mutants of CD163 were performed to confirm that scavenger receptor cysteine-rich domain 5–9 (SRCR5–9) domain was an interaction site for the virus, which was critical for virus infection [30], while the other four SRCRs at the N-terminal and the cytoplasmic tail were not required for CD163 to sustain PRRSV infection [24].

Here, to pave the way for CD163 targeted antiviral strategies against PRRSV, several mAbs of SRCR5–9 were screened for PRRSV suppression. Among the candidates, 6E8 and 9A10 presented broad inhibition activity against different PRRSV strains in PAMs and Marc-145 cells either pre- or post-attachment. Besides, the treatment with either 6E8 or 9A10 leads to the suppression in CD163 transcription. Further, the novel residues ^570^SXDVGXV^576^ and Q^797^ recognized by 6E8 and 9A10 individually were identified to play a role in PRRSV infection. These findings create a deeper understanding in CD163 function and regulation related to PRRSV.

## 2. Materials and Methods

### 2.1. Cell Culture and Viruses

Marc-145 cells (African green monkey kidney cells) and HEK293T cells were cultured in Dulbecco’s minimal essential media (DMEM, Hyclone, Thermo Scientific, MA, USA) with 10% fetal bovine serum (FBS, Invitrogen, USA) at 37 °C in 5% CO_2_. Porcine alveolar macrophages (PAMs) were prepared as described previously [31]. PRRSV strains including ZJfh17 (lineage 8), HNxx16 (lineage 1), ZJnb16–2 (lineage 8, 3), ZJhz16–2 (lineage 5), and JS18–3 (lineage 8) were previously isolated in CATG lab, Zhejiang University, China and titrated in PAMs. Commercially modified-live PRRS vaccines with corresponding virus strains named HuN4-F112 (Harbin Weike Biotechnology Development Company, Harbin, China) were propagated and titrated in Marc-145 cells. Sf9 insect cells were used to propagate recombinant baculoviruses in SF900 III SFM (Invitrogen, Carlsbad, CA, USA) at 27.5 °C.

### 2.2. Expression of CD163 SRCR5–9

Truncated CD163 SRCR5–9 fragments with a honeybee melittin signal peptide were inserted into *Eco*RI and *Xho*I sites of the transfer vector pFBD. Recombinant baculoviruses were subsequently generated using Bac-to-Bac system (Invitrogen, Carlsbad, CA, USA), according to the manufacturer’s instructions. Sf9 insect cells were cultured in SF900 III SFM and then inoculated with recombinant baculoviruses at a multiplicity of infection (MOI) of 1. At 96 h post infection, the culture was centrifuged at 12,000 rpm for 30 min, and protein was purified using Ni-NTA affinity column (Novagen, Madison, WI, USA) as described previously [32].

### 2.3. SDS-PAGE and Western Blotting

SDS-PAGE and Western blotting were performed according to standard protocols [33]. Briefly, protein samples were separated on 12% SDS-PAGE gels and transferred to polyvinylidene fluoride (PVDF) membranes (Merck Millipore, Billica, USA). Membranes were blocked with 5% (w/v) nonfat milk in PBS containing 0.05% Tween (PBST) for 1 h at 37 °C. Rinsed blots were incubated with anti-His monoclonal antibody (1:5000 in PBS, Abcam, Cambridge, UK), CD163, or PRRSV-specific monoclonal antibodies (prepared in our lab) at 37 °C for 1 h (h) respectively, and with HRP-conjugated goat anti-mouse IgG (1:5000 in PBS, Sungene, Shanghai, China) at 37 °C for 1 h. Signals were detected by SuperSignal West Pico/Femto Chemiluminescent Substrate (Thermo Scientific, MA, USA) and images were captured with Gel 3100 chemiluminescent imaging system (Sage Creation Science, Beijing, China).

### 2.4. Production and Characterization of Murine mAbs

Monoclonal antibody production was performed as described previously [34]. Briefly, BALB/c mice were subcutaneously immunized twice in a 2-week interval with recombinant CD163 SRCR5–9 in ISA-206 adjuvant (Seppic, Paris, France). Mice were boosted with the same dose of antigen 3 days before the fusion of splenocytes and SP2/0 cells. Hybridoma was screened in IFA. Positive hybridomas were cloned by limiting dilution, expanded, and further subcultured. Culture supernatant was clarified and tested for inhibition assay as described below. Immunoglobulins from selected positive mAbs were isotyped using a commercial isotyping kit (Amersham Bioscience, Piscataway, NJ, USA) according to the manufacturer’s protocol. mAbs were purified with Montage kit Prosep-G (Millipore, Billica, MA, USA) for IgG. All experiments involving animals were performed in biosafety level 2 animal facilities in accordance with the institutional biosafety manual.

### 2.5. Immunofluorescence Assays

Cells in 96-well plates (Costar; Corning, NY, USA) were infected with viruses at MOI of 1. At 48 h post infection, cells were fixed with 80% ice-cold acetone in PBS for 30 min at −20 °C, washed twice in PBS, and blocked with 5% (w/v) non-fat milk in PBS for 1 h. Primary antibodies diluted in PBS containing 0.5% non-fat milk were incubated for 1 h at 37 °C and then washed with PBS for three times. Cells were incubated with FITC-conjugated goat anti-mouse IgG (H+L) (1: 1000 in PBS, Thermo Scientific, Carlsbad, CA, USA) as secondary antibody at 37 °C for 1 h. Finally, labeled cells were treated with nuclear dye 4′,6′-diamidino-2-phenylindole dihydrochloride (DAPI, 1: 2000 dilution in PBS, Beyotime, Shanghai, China). Cells were examined with an inverted fluorescence microscope (Olympus, Corporation, Tokyo, Japan).

### 2.6. Pre-Attachment Assay

PAMs or Marc-145 cells grown in 24-well plates were incubated with mAb against CD163 at various concentrations at 37 °C for 1 h. Following incubation, cells were washed three times with PBS and inoculated with PRRSV (100 TCID_50_) for 1 h at 37 °C. Unbound virus was removed by washing three times with PBS. At 24 h post inoculation, cells was collected and tested by IFA, Western blotting, and qRT-PCR.

### 2.7. Post-Attachment Assay

First, 100 TCID_50_ of PRRSV was added to PAMs in a 24-well plate, and then the plate was incubated at 37 °C for 1 h to allow virus attachment. The plate was thoroughly washed for three times. Subsequently, 100 μg/mL mAbs were added to the wells and incubated at 37 °C for 1 h. After the plate was rinsed three times, cells were incubated at 37 °C for 24 h. Virus expression was evaluated in qRT-PCR.

### 2.8. Construction of Stable Cell Lines Expressing CD163

CD163 mutants were cloned into lentivirus vector pCD513B-1 (SBI, Mountain View, CA, USA), according to the manufacturer’s instructions. HEK293T cells grown in 10-cm dishes were co-transfected with the recombinant plasmid and three helper plasmids (pGag/Pol, pRev, pVSVG), as described by the manufacturer. At 48 h post infection (hpi), virus supernatant was harvested concentrated as described previously [35]. The concentrated virus was inoculated into 3D4 in 6-well plates. Expression of CD163 mutants tagged with enhanced GFP (EGFP) in 3D4 was tested in IFA and Western blotting.

### 2.9. RNA Isolation and qRT-PCR

Total RNA was extracted from uninfected and virus-infected cells using TRIzol (Tiangen, Beijing, China) according to the manufacturer’s instructions and transcribed to cDNA by using HiScript III 1st Strand cDNA Synthesis Kit (+gDNA wiper) (Vazyme, Nanjing, China). qRT-PCR was performed on Stratagene Mx3005P real-time PCR system (Agilent Technologies, Santa Clara, CA, USA) with SYBR green PCR mix according to the manufacturer’s instructions (Vazyme). GAPDH gene was used as an internal control. Specificity was confirmed by sequencing of the PCR products and melting-curve analysis for qPCR. Relative mRNA expressions of N, CD163, NF-KB, IL-1β, and IL-8 were detected using qRT-PCR and calculated using the 2-ΔΔ Ct method [36]. Three replicates were included for each treatment. All primers for RT-PCR are listed in Table 1.

### 2.10. Statistical Analysis

Data were expressed as the mean ± SD of three independent experiments. Statistical significance was calculated using a one-way analysis of variance (ANOVA) with multiple comparisons in GraphPad Prism 5 (GraphPad Software, San Diego, CA, USA). Asterisks *, **, *** or **** in figures indicate statistical significance at the *p* < 0.05, *p* < 0.01, *p* < 0.001, or *p* < 0.0001 level, respectively.

## 3. Results

### 3.1. mAbs 6E8 and 9A10 Were Selected for Specific Recognition of CD163

CD163 is a scavenger receptor for PRRSV, of which SRCR 5–9 has been shown to be required for PRRSV infection in vitro. To evaluate the function of SRCR5–9 in PRRSV replication, purified SRCR5–9 was incubated with the virus before the inoculation to PAMs. As shown in Figure 1A, PRRSV infection was significantly inhibited with SRCR5–9 in PAMs, indicating the successful PRRSV blocking by SRCR5–9. Further, CD163 mAbs were produced and screened. Two mAbs (6E8 and 9A10) were selected based on the specific recognition of both native and recombinant CD163 (Figure 1C,D). Neither 6E8 nor 9A10 reacts with CD163 SRCR 5–9 in Western blot, suggesting that the conformational epitopes were recognized by two mAbs, instead of linear epitopes (Figure 1B).

### 3.2. 6E8 and 9A10 Significantly Block PRRSV Infection

6E8 and 9A10 were identified as IgG1 and purified. To verify antiviral activity of mAbs against PRRSV, virus inhibition assay was primarily performed in PAMs. Viral expression of PRRSV ZJfh17 was significantly controlled after incubation with mAbs 6E8 and 9A10 at a concentration of 50 μg/mL. As shown in Western blot, virus titration, and qRT-PCR, the inhibition efficiency of 6E8 on ZJfh17 was higher than 9A10 (Figure 2A–C). Further, 6E8 and 9A10 displayed an inhibitory effect on ZJfh17 infection in a dose-dependent manner and 400 μg/mL mAbs completely inhibited the infection of ZJfh17 in PAMs, while no inhibition was observed with PRRSV irrelevant mAb 6D10 at the same dose (Figure 2D). In addition, the inhibition efficiency of two mAbs mixed at a 1:1 ratio was higher than either antibody at the same total antibody concentration (Figure 2E), suggesting that 6E8 and 9A10 target different epitopes in CD163 and provide complementary inhibition against PRRSV. Taken together, these results demonstrated that 6E8 and 9A10 exhibit effective PRRSV inhibition via CD163.

### 3.3. 6E8 and 9A10 Provide Broad Inhibition Against Various PRRSV Strains

Since CD163 was identified as the common receptor for PRRSV, the potential of CD163 antibodies in universal protection was evaluated with different PRRSV strains. As shown in Figure 3, 6E8 and 9A10 exhibited effective inhibition on variable representative strains of different PRRSV genotypes in PAMs and Marc-145, especially, ZJfh17 (lineage 8), ZJnb16–2 (lineage 8, 3), ZJhz16–2 (lineage 5), and HuN4 -F112 (lineage 1). 6E8 displays more effective inhibition on PRRSV strains as compared to 9A10. These results confirm that mAbs against CD163 SRCR 5–9, especially 6E8, display significantly broad anti-PRRSV activity in PAMs and Marc-145.

### 3.4. 6E8 and 9A10 Stop PRRSV Infection Either Before or After Virus Attachment

To explore the function of CD163 mAbs for PRRSV prevention and treatment, the inhibition with mAbs was evaluated pre- and post-attachment in PAMs. mAbs were incubated with cells either before or after virus attachment. As shown in Figure 4A, 6E8 and 9A10 at a concentration of 100 μg/mL efficiently block CD163 receptor before virus attachment and prevent PRRSV infection. Meanwhile, after the incubation of virus at 37 °C for 1 h, antibodies were added to the culture at a concentration of 100 μg/mL. 6E8 and 9A10 successfully stop further infection steps post-virus-attachment (Figure 4E). The inhibition efficiency of 6E8 is higher than that of 9A10 in both tests. The results indicated that mAbs 6E8 and 9A10 inhibit PRRSV infection via receptor blocking either pre- or post-attachment, suggesting their application potential in both PRRSV prophylactics and therapeutics.

PRRSV replication triggers the activation of NF-κB, ERK1/2, and p38MAPKs, leading to the release of proinflammatory cytokines such as IL-1β, IL-8, and IL-6. As shown in Figure 4, mRNA expression of NF-κB, IL-1β, and IL-8 significantly decreased with either pre- or post-attachment antibody treatment at 100 μg/mL individually. The results confirmed that CD163 mAbs (6E8 and 9A10) effectively inhibit the PRRSV related NF-κB pathway and prevent cellular pathogenesis.

### 3.5. 6E8 and 9A10 Suppress CD163 Transcription in PAMs and Marc-145

To further elucidate the function of CD163 antibody treatment in target cells, regulation in CD163 transcription was evaluated. Upon the treatment with 6E8 and 9A10 individually, significant decrease was observed in CD163 mRNA level in PAMs (Figure 5A) and Marc-145 (Figure 5B). The inhibition efficiency with 6E8 was more significant as compared to 9A10. As membrane associated CD163 is subjected to the cleavage into secreting CD163, specific antibody binding may interfere with CD163 cleavage or degradation due to the competition between antibody and protease. Marc-145 was treated with MG132, a protease inhibitor, to stop all the possible CD163 cleavage or degradation. Suppression in CD163 transcription was observed with MG132 treatment (Figure 5C). As shown in Figure 5D, the treatment with either MG132 or mAbs for 12 to 24 h leads to the similar decline in CD163 transcription. These results provide another mechanism for CD163 targeted antibody therapy against PRRSV and suggest that 6E8 and 9A10 may compete not only with PRRSV but also with proteases for CD163 interaction.

### 3.6. Epitope Mapping for 6E8 and 9A10 in CD163

To determine the minimal antigenic domains recognized by 6E8 and 9A10, various truncated CD163 SRCR5–9 proteins were designed and expressed in baculovirus. Subsequently, the immunoreactivity between the two mAbs and these fragments were screened in IFA. 6E8 reacted with fragments spanning F2, but not to F3 to F5, suggesting that the epitope was located around aa90-aa101 (Figure 6A). Meanwhile, epitope recognized by 9A10 was mapped to aa305-aa331 (Figure 6C). To further precisely define the minimal motifs required for antibody binding, a series of point mutations were introduced to CD163 SRCR5–9 individually. Mutated SRCR5–9s were screened with the corresponding mAb in IFA. Substitutions S94P, D96P, V97P, G98P, and V100P lead to the abolishment of immunofluorescence (green) signals upon 6E8 immunostaining, indicating an epitope consisting of five critical residues, ^570^SXDVGXV^576^ (Figure 6B). Similarly, the epitope of 9A10 was identified to residues Q^797^ (Figure 6D).

With CD163 structure analysis, ^570^SXDVGXV^576^ and Q^797^ were found to be located on the surface of CD163, which facilitates the interaction with either virus RBDs or antibodies (Figure 7A). As shown in sequence alignment, the two epitopes identified are conserved among different species, including pigs, humans, and monkeys (Figure 7B), suggesting the key roles of these residues in maintaining CD163 function.

### 3.7. *Residues*
^570^SXDVGXV^576^ and Q^797^ of CD163 Are Required for PRRSV Infection

To further study the function of residues recognized by mAbs in PRRSV infection, CD163s with or without mutated epitopes were overexpressed in 3D4 cells, in which endogenous CD163 expression is absent. Specifically, 3D4 cell lines designated 3D4^CD163^, 3D4^CD163−6E8-mutant^, and 3D4^CD163−9A10-mutant^ were generated via infection with recombinant lentiviruses encoding porcine full-length CD163, CD163–6E8-mutant and CD163–9A10-mutant, respectively. First, expression of CD163 and mutants in 3D4 was confirmed by IFA (Figure 8A). 3D4^CD163^, 3D4^CD163−6E8-mutant^, and 3D4^CD163−9A10-mutant^ were infected with PRRSV at MOI of 1, while wild type 3D4 served as control. As shown in IFA, 3D4^CD163^ supported PRRSV replication while original 3D4 failed to. Notably, upon PRRSV inoculation, only sporadic cells were infected in 3D4^CD163−9A10-mutant^, while no infection was detected in 3D4^CD163−6E8-mutant^. In Western blotting, PRRSV infection were observed in 3D4^CD163^ at 24 h post-infection, while no virus specific signal was detected in either 3D4^CD163-mutants^ or wild type 3D4 (Figure 8B). Similarly, in qPCR, no PRRSV replication was observed in 3D4^CD163−6E8-mutant^, while robust viral signals were obtained from 3D4^CD163^ (Figure 8C). Slight PRRSV expression was detected in 3D4^CD163−9A10-mutant^, indicating any substitution in epitopes of 6E8 and 9A10 abolishes or significantly hampers PRRSV infection. For the first time, the key residues in CD163 for PRRSV infection have been identified.

## 4. Discussion

From the first identification, PRRSV has constantly threatened the global pig industry with continuing evolving virulent PRRSV strains for several decades. The typical immune features of PRRSV infection in host include persistence viremia, a strong inhibition of innate cytokines, dysregulation of NK cell function, rapid induction of non-neutralizing antibodies, delayed appearance of neutralizing antibody, a late and low CD8+ T-cell response, and induction of regulatory T cells [37]. To prevent PRRSV infection, modified live and inactivated PRRS vaccines have been developed, but there are concerns about the biosafety and efficacy, such as viral shedding, recombination with field strains, reversion to virulence, and failure to trigger a protective immune response against heterogeneous isolates [38]. Hence, it is necessary to develop a broad antiviral strategy for infected herds. Receptor targeted antibody therapy is emerging as one alternative option for universal virus inhibition.

In comparison to conventional vaccines, direct disruption in the interaction between host receptor and virus is one of the most effective antiviral strategies. As most viral receptors in host are conserved, these receptor targeted alternatives are functional against variable strains. Besides, the interaction to receptors is the initial of the infection. Hence, receptor blocking strategies completely prevent virus entry as compared to those infection-permissive intracellular inhibitors. The receptor-binding domain (RBD) of N-acetyl-neuraminic acid receptor is highly conserved in different HA subtypes and can be targeted for the development of broad-spectrum antivirals to block the receptor binding of HA, including antibody, receptor-binding, and destroying agents [39]. Besides, RBD on COVID-19 S protein can bind specifically to ACE2 receptor, resulting in RBD recombinant protein and ACE2 receptor blocker (chloroquine, antibody) can be used as candidate vaccines for COVID-19 treatment [40]. Moreover, mAbs have long been an integral tool in basic research due to their high specificity and affinity for target antigens. Therefore, CD163 receptor is the preferred target to develop a broad antiviral antibody to block the binding of the receptor and PRRSV. Numerous studies have demonstrated that CD163 is an indispensable cellular receptor for PRRSV infection [24,41]. Of the nine SRCR domains within the extracellular region (SRCR1–9), SRCR5–9 Fc protein of CD163 shows an additive anti-PRRSV activity due to its binding to PRRSV virions [42] and SRCR5 domain is particularly crucial and indispensable for successful PRRSV infection [43,44]. However, purified SRCR5 domain fails to capture PRRSV virions, indicating SRCR5 is not sufficient to block the virus. In this study, 6E8 and 9A10 against SRCR5–9 were selected based on the significant activity to inhibit PRRSV strains infection in a dose-dependent manner in PAMs and Marc-145 (Figure 2). Here, 6E8 and 9A10 displayed strong anti-PRRSV activity against different PRRSV lineages, especially lineage 8 and 5. These findings confirmed that CD163 blocking with antibodies exhibited broad antiviral effects against PRRSV.

Meanwhile, as noticed that 6E8 and 9A10 shown variant inhibitory efficiency against different PRRSV strains, this suggests that PRRSVs with different genetic features in GP2a and GP4, the major CD163 targeted structural proteins [45], have difference in the dependence on CD163 in virus entry. Previously, genetically modified pigs possessing SRCR 5 substitution with domain homologs from human CD163L1 were not permissive for genotype 1 PRRSV infection but susceptible to the infection by genotype 2 viruses [44]. Besides, similar results about the efficacy difference against variable strains were noted in another study, which established genetically modified pigs with deletion of the SRCR 5 domain of CD163 [23]. Therefore, it is believed that the disruption in different regions of CD163 will lead to different inhibitory effects on different PRRSV strains, implying CD163 may play multiple roles in PRRSV infection.

To further explore the function of CD163 mAbs for PRRSV prevention and treatment, the inhibition with the antibodies was evaluated pre- and post-attachment in PAMs. 6E8 and 9A10 have significantly preventive and therapeutic effects in vitro. Both antibodies successfully stop the virus infection cycle even after virus attachment completes. As known to us, PRRSV arouses NF-κB activation and leads to the release of several cytokines and the TLR4/MyD88/NF-κB signaling pathway for IL-1β production [46,47]. Meanwhile, as shown in Figure 3, 6E8 and 9A10 also antagonize PRRSV-caused pathogenicity in cells, including the inhibition of the inflammatory factor. The difference in inhibitory efficiency in IL8 and IL-1β between pre- and post-attachment implies virus attachment may have different effects on innate immunity and antibodies targeting different epitopes may contribute to the process differently. These results verified that 6E8 and 9A10 presents significant antiviral activity, leading to a promising application in therapeutics and prevention.

As a type I membrane protein, the extracellular portion of CD163 is anchored to the cell surface by a single transmembrane region with a short cytoplasmic tail. CD163 is cleaved from the cell membrane by a proteolytic shedding mechanism in response to PMA, activation of Toll-like receptors such as Fcγ receptor cross-linking, and inflammatory stimuli such as lipopolysaccharide; the cleavage produces soluble CD163 (sCD163) [30,48]. Interestingly, 6E8 and 9A10 lead to transcription suppression of CD163 in PAMs and Marc-145, which was consistent with the action of MG132 protease inhibitor (Figure 4). These results suggested that 6E8 and 9A10 can block proteolytic modification on CD163, leading to the accumulation of intact CD163 and the suppression of CD163 transcription consequently. As the upregulation of CD163 facilitates PRRSV entry, the minimized CD163 expression will further hinder the infection. Hence, 6E8 and 9A10 mediate PRRSV inhibition not only via blocking the existing receptor but also by the suppression of the functional CD163 production.

Currently, limited detail about CD163 and virus interaction is available. The key residues involved in the virus attachment and internalization have not been identified previously, hindering the understanding in virus pathogenesis and the antiviral development [49]. Here, epitope mapping was performed for the CD163 mAbs. ^570^SXDVGXV^576^ for 6E8 and Q^797^ for 9A10 were identified to be indispensable for antibody recognition. Moreover, in CD163-chimeric 3D4 cells, epitope mutants fail to support PRRSV infection, indicating these residues are also required for virus and host interaction. Taken together, these results demonstrated that residues ^570^SXDVGXV^576^ and Q^797^ serve as the novel target in CD163 to develop antiviral agents.

In summary, studies delivered that mAbs 6E8 and 9A10, targeting the host receptor, present broad antiviral activity against variable PRRSV strains, showing advantages over conventional virus-targeted strategies. Both mAbs exhibited the potential for PRRSV prevention and treatment in vitro. Notably, the study revealed that the virus inhibitory function of the antibodies is resulted from not only the receptor blocking but also the suppression in receptor transcription. This indicated the complexity in antibody-mediated host protein regulation: the simultaneous effects on both virus and host. Fortunately, in the study, the effects from 6E8 and 9A10 on both virus and host contribute to the virus inhibition. Moreover, with the antibodies, two novel epitopes, ^570^SXDVGXV^576^ and Q^797^, were identified in CD163, which play a vital role in PRRSV infection. This finding will help to further unveil the interaction mechanisms between PRRSV and CD163. Further efforts will be made to apply CD163-antibody therapy in vivo, in order to benefit the industry suffered from the disease. Functional recombinant Fabs or CDRs from the antibodies will be produced in cost-effective platforms for veterinary purpose. Meanwhile, a delivery strategy and immunization schedule will be explored to fit pig production and swine disease control procedures. Overall, this study on CD163 mAbs provides an alternative universal antiviral candidate against PRRSV and throws light on CD163 additional function in PRRSV regulation.

## Figures and Tables

**Figure 1 vaccines-08-00592-f001:**
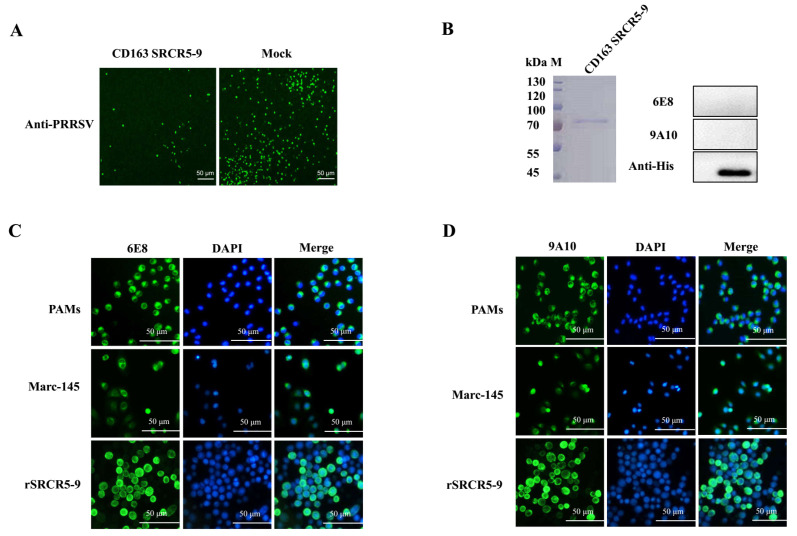
Production and characterization of 6E8 and 9A10 against CD163 SRCR5–9. (**A**) CD163 SRCR5–9 protein inhibited PRRSV infection in PAMs. CD163 SRCR5–9 was purified by Ni-NTA Agarose. Then, 100 μg/mL CD163 SRCR5–9 was preincubated with 100 TCID50 ZJfh17 for 1 h and further inoculated in PAMs. (**B**) 6E8 and 9A10 showed no reactivity with purified CD163 SRCR5–9 protein in Western blotting. Anti His-tag mAb was used as the control. (**C**,**D**) Native CD163 in PAMs, Marc-145, and recombinant CD163 SRCR5–9 in High Five cells were detected by 6E8 and 9A10. Cells were co-immunostained with DAPI, and probed with Alexa-Fluor-labeled secondary antibodies. rSRCR5–9: recombinant CD163 SRCR5–9 expressing in SF9 cells.

**Figure 2 vaccines-08-00592-f002:**
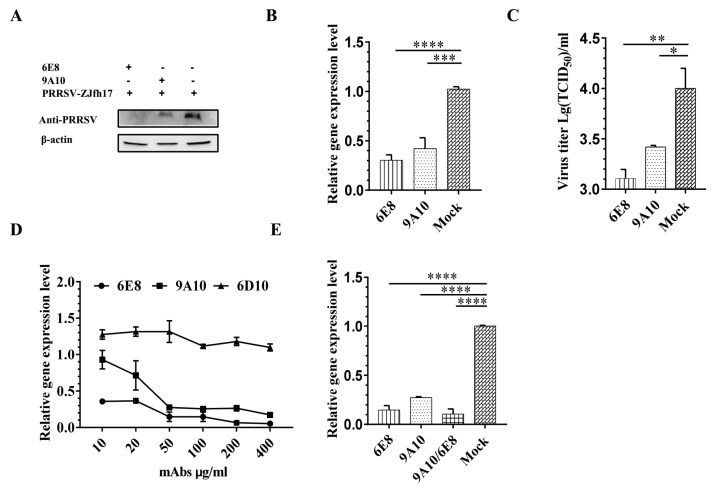
6E8 and 9A10 block PRRSV infection in PAMs. Inhibition in PRRSV infection by 6E8 and 9A10 in PAMs was detected using Western blotting (**A**), qRT-PCR (**B**), and virus titration (**C**). PAMs were preincubated with 50 μg/mL mAb for 1h, followed by the inoculation with PRRSV ZJfh17. Cells were collected at 24 hpi and titrated. β-actin was detected as the internal controls. (**D**) Virus inhibition with 6E8 and 9A10 at different concentrations in PAMs. 6D10 served as an irrelevant mAb control. (**E**) Comparison between mAbs (6E8, 9A10, and 9A10+6E8) in ZJfh17 inhibition in PAMs. 9A10 and 6E8 were mixed at a ratio of 50:50 (w/w). PRRSV N expression was used to indicate relative changes of viral mRNA as compared to the infection group without antibody treatment after normalization using GAPDH mRNA. Experiments were performed in triplicate and data were shown as mean ± SD. Statistical significance is indicated as * (*p* < 0.05); ** (*p* < 0.01); *** (*p* < 0.001); **** (*p* < 0.0001).

**Figure 3 vaccines-08-00592-f003:**
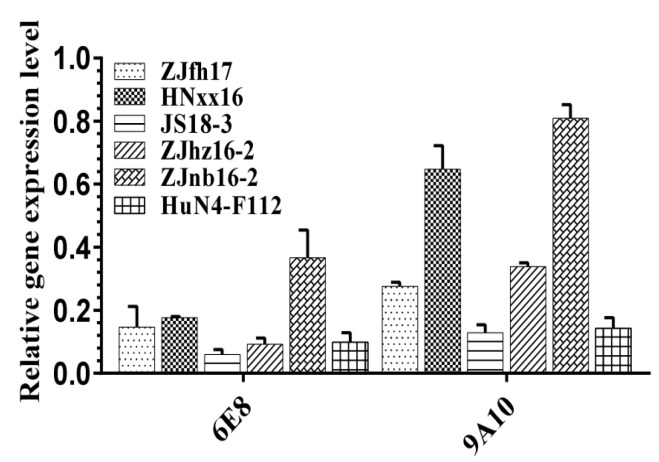
6E8 and 9A10 inhibit different PRRSV strains. PAMs and Marc-145 were preincubated with 50 μg/mL mAb for 1 h, followed by the inoculation with PRRSV strains individually, including ZJfh17, HNxx16, JS18–3, ZJnb16–2, ZJhz16–2, and HuN4-F112. Cells were collected at 24 hpi and relative PRRSV N mRNA levels were quantified by real-time PCR. Relative changes of mRNA were compared to the PRRSV infection group without antibody treatment. GAPDH was detected as the internal reference.

**Figure 4 vaccines-08-00592-f004:**
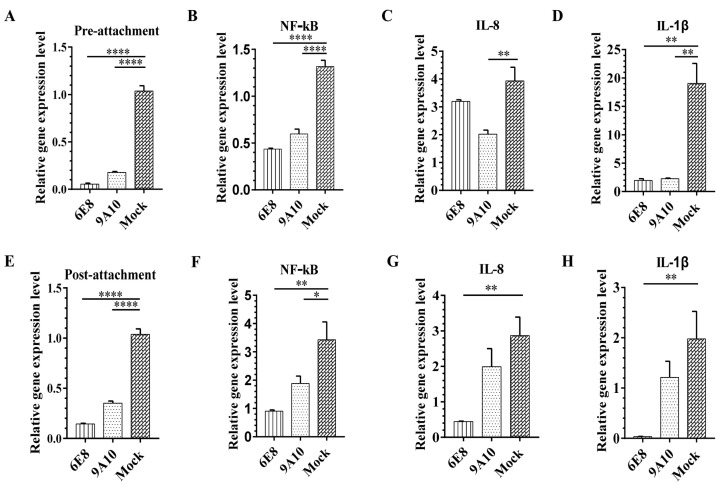
9A10 and 6E8 inhibit infection and PRRSV-related NF-κB pathway pre- and post-attachment. For pre-attachment assays, PAMs were pre-incubated with mAbs (100 μg/mL) at 37 °C for 1 h, and inoculated with 100 TCID50 ZJfh17 for 1 h. For post-attachment assays, PAMs were treated with ZJfh17, and then post-incubated with mAbs. After 24 hpi, total RNA was extracted and used as template for qRT-PCR using primers specific for N (**A**,**E**), NF-κB (**B**,**F**), IL-8 (**C**,**G**), and IL-1β (**D**,**H**). GAPDH was detected as the internal reference. Experiments were performed in triplicate and data are shown as mean ± SD. Statistical significance is indicated as * (*p* < 0.05); ** (*p* < 0.01); **** (*p* < 0.0001).

**Figure 5 vaccines-08-00592-f005:**
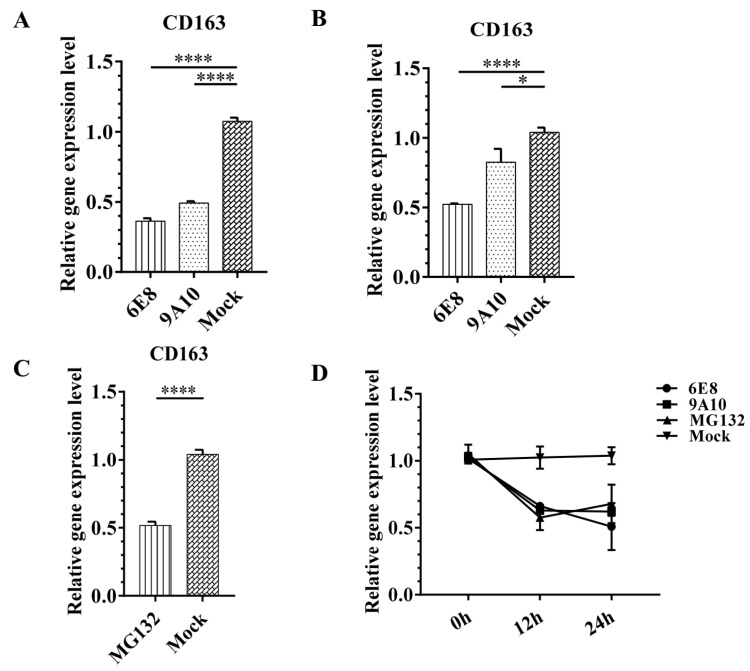
9A10 and 6E8 mAbs suppress CD163 transcription. Total RNA was extracted to evaluate mRNA expression of CD163 in PAMs (**A**) and Marc-145 (**B**). Cells were mock treated or treated with mAbs (100 μg/mL) for 1 h, and then cultured for 24 h. (**C**) MG132 (20 μg/mL) inhibited CD163 transcription in Marc-145. (**D**) Time-coursed suppression in CD163 transcription with mAbs or MG132. GAPDH was detected as the internal reference. Experiments were performed in triplicate and data are shown as mean ± SD. Statistical significance is indicated as * (*p* < 0.05); **** (*p* < 0.0001).

**Figure 6 vaccines-08-00592-f006:**
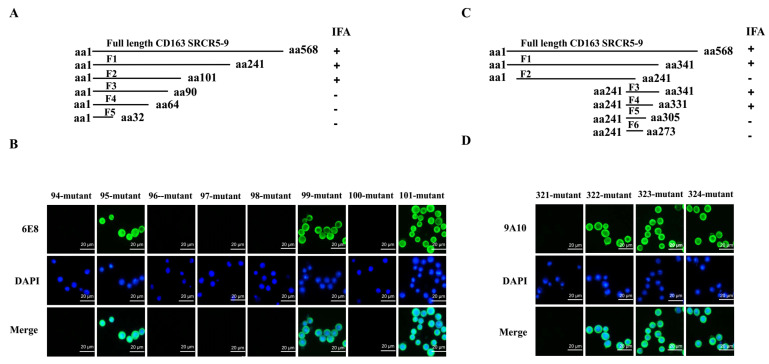
Fine mapping of epitopes recognized by 6E8 and 9A10. Schematic diagram of various CD163 SRCR5–9 truncated fragments for epitope mapping with 6E8 (**A**) and 9A10 (**C**). SRCR5–9 fragments 1 to 5 and subfragments were expressed in recombinant baculovirus in Sf9 insect cells and screened with individual mAb in IFA. (**B**,**D**) IFA with mAbs in insect cells expressing point mutated Ac-CD163 SRCR5–9. Alanine substitutions at aa90-aa101 (**B**) and aa305-aa331 (**D**) of Ac-CD163 SRCR5–9 were performed individually.

**Figure 7 vaccines-08-00592-f007:**
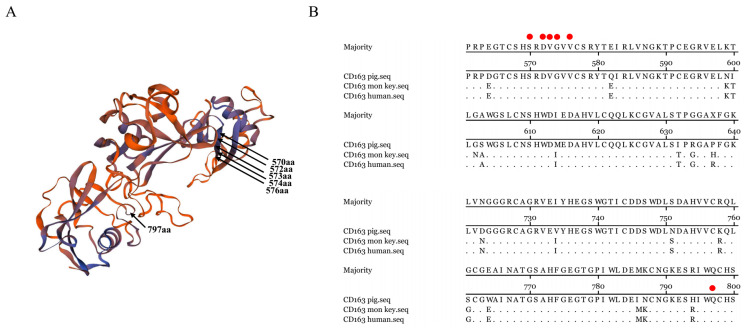
Structure and sequence analysis of CD163. (**A**) Structure prediction of CD163 by software, where the solid arrows indicate residues of mAbs (6E8 and 9A10) (**B**) Sequence alignment of CD163 from different species. Amino acid sequences of CD163 from pig (p; UniProt entry Q2VL90), human (h; UniProt entry Q86VB7), and monkey (M; UniProt entry Q2VLG4) are aligned. The epitope related residues were indicated with red dots.

**Figure 8 vaccines-08-00592-f008:**
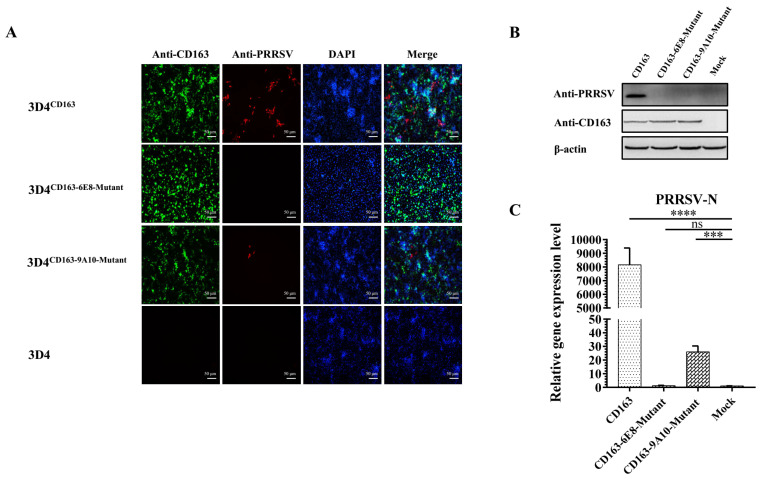
Epitopes of 6E8 and 9A10 in CD163 are required for PRRSV infection. 3D4CD163 and 3D4 CD163-mutants cell lines were infected with PRRSV ZJfh for 24 h. (**A**) Cell lines were subjected to immunofluorescence staining for PRRSV (red), CD163 (green), and cellular nuclei (blue, DAPI). Co-localization of CD163 and PRRSV was visualized by fluorescent microscopy. (**B**) Western blotting for PRRSV in 3D4CD163 and 3D4 CD163-mutants cell lines. β-actin was detected as the internal controls. (**C**) PRRSV mRNA expression in 3D4CD163 and 3D4 CD163-mutants cell lines. Total RNA was extracted and used as template for qRT-PCR using primers specific for PRRSV N. GAPDH was detected as the internal reference. Data are the results of three independent experiments (means ± SD). Significant differences are denoted by *** (*p* < 0.001); **** (*p* < 0.0001); ns, not significant.

**Table 1 vaccines-08-00592-t001:** List of primers for RT-PCR.

Prmier	Sequence(5′–3′)
ORF(N)-F	AAAACCAGTCCAGAGGCAAG
ORF(N)-R	CGGATCAGACGCACAGTATG
CD163-F	ATTCATCATCCTCGGACCCAT
CD163-R	CCCAGCACAACGACCACCT
NF-κB-F	TCGCTGCCAAAGAAGGACAT
NF-κB-R	AGCGTTCAGACCTTCACCGT
IL-1β-F	CCCAAAAGTTACCCGAAGAGG
IL-1β-R	TCTGCTTGAGAGGTGCTGATG
IL-8-F	GGCATCACCTTTGGCATCTT
IL-8-R	TGGCATCGAAGTTCTGCACT
mGAPDH-F	TGACAACAGCCTCAAGATCG
mGAPDH-R	GTCTTCTGGGTGGCAGTGAT

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
