# Peer review of "CD163 Antibodies Inhibit PRRSV Infection via Receptor Blocking and Transcription Suppression"

_vaccines, 2020, doi:10.3390/vaccines8040592_

Round 1

Reviewer 1 Report

The manuscript submitted by Xu et al., entitled "CD163 antibodies inhibit PRRSV infection via receptor blocking and transcription suppression" aims to demonstrate that two antibodies can be an efficient therapy against PPRS, as well as, to clarify the role of the CD163 receptor in the infection. The used methods are accurate and robust, and results are interesting to the field.

The following corrections/suggestions should be performed before the manuscript's acceptance: 

Line 55 - please define the origin of the Marc-145 cell line;

Line 111 - An ethical statement should be added

Figure 2A - Please provide the image of all immunoblot membrane (in the figure or as supplementary data)

Figure 2B, D, E - Please clarify the viral genes that were quantified

Figure 3 - Justify the discordant differences between the two antibodies among different PPRS serotypes

Figure 4 - please justify the apparent discordant results IL-8 and IL-1Beta of both antibodies between pre- and post-attachment exposures

In the discussion section, the manuscript will be improved if the authors discuss in deep the future practical applications of immunotherapy against CD164 in pig.

Author Response

Editor of Vaccines

Dear Editor,

Thank you very much for conveying the reviewers’ reports. Please find enclosed the revised manuscript entitled “CD163 antibodies inhibit PRRSV infection via receptor blocking and transcription suppression” by Huiling Xu, Zehui Liu, Suya Zheng, Guangwei Han, Fang He *.

We found the comments raised by the reviewers to be helpful for improving the quality of our manuscript, and have responded accordingly with additional analysis and proofreading. Point-by-point responses are attached below for your perusal. We trust that you find our revised manuscript better suited for publication in Vaccines than our original submission, and we look forward to your favorable response.

Yours sincerely,

HE Fang

Principal Investigator

Institute of Preventive Veterinary Medicine

College of Animal Sciences

Zhejiang University

866 Yuhangtang Road, Hangzhou

China

-------------------------------------------------------------------------------------------

Reviewer #1 (minor concerns):

“Line 55 - please define the origin of the Marc-145 cell line”

Page 2, Line 56---------Information has been provided accordingly.

“Line 111 - An ethical statement should be added”

Page 3, Line 114-116--------- Information has been provided accordingly.

“Figure 2A - Please provide the image of all immunoblot membrane (in the figure or as supplementary data)”

Figure 2A ---------- The whole image of immunoblot membrane and the description have been presented in supplementary data.

“Figure 2B, D, E - Please clarify the viral genes that were quantified”

Page 6, Line 203-205---------Clarification has been made accordingly.

“Figure 3 - Justify the discordant differences between the two antibodies among different PPRS serotypes”

Page 12, Line 347-369---------Clarification has been made in Discussion accordingly.

In brief, 6E8 and 9A10 recognize different epitopes, 570SXDVGXV576 in SRCR5 and Q797 in SRCR7. The disruption in different regions of CD163 will lead to different inhibitory effects on different PRRSV strains due to the different dependence on CD163.

“Figure 4 - please justify the apparent discordant results IL-8 and IL-1Beta of both antibodies between pre- and post-attachment exposures”

Page 12, Line 376-379------Clarification has been made in Discussion accordingly.

“In the discussion section, the manuscript will be improved if the authors discuss in deep the future practical applications of immunotherapy against CD164 in pig.”

Page 13, Line 410-414---------Application of the antibodies has been discussed accordingly.

Reviewer 2 Report

This is an interesting article describing mAbs blocking PRRSV infection.

PRRSV is one of the major threat for the pig industry worldwide. Two mAbs (6E8 and 9A10) against SRCR5-9 were shown to be able to block PRRSV infection in a dose-dependent manner. They also inhibited PRRSV-related NF-kb pathway, with inhibition of IL-8 and IL-1beta gene expression. In addition, two novel epitopes were identified within CD163: both 570SXDVGXV576 and Q797 residues were crucial in PRRSV infection

Introduction was well-focused, matherials and methods were described in details, and results clearly illustrated and discussion is well written and well structured.

I personally think that this study is devoid of major weakness and only minor corrections are need to improve it.

Line 91. Introduce 'min'.

Line 99. Introduce 'h'.

Line 118. Substitute 2000 with '2,000'.

Line 138. Introduce 'hpi'

Line 149. Remove the space within Δ.

Line 170. Please add also in the figure legend for how many hours did you pre-incubate CD163 SRCR5-9 with PRRSV.

Figure 6. In which cell lines did you perform IFA?

Discussion. Although I find it well written and easy to ready, I think that it should be highlightened a bit more that this is only an in vitro study, and more study in vivo are need to assess the real applicability of CD163-antibody therapy against PRRSV.

Author Response

Editor of Vaccines

Dear Editor,

Thank you very much for conveying the reviewers’ reports. Please find enclosed the revised manuscript entitled “CD163 antibodies inhibit PRRSV infection via receptor blocking and transcription suppression” by Huiling Xu, Zehui Liu, Suya Zheng, Guangwei Han, Fang He *.

We found the comments raised by the reviewers to be helpful for improving the quality of our manuscript, and have responded accordingly with additional analysis and proofreading. Point-by-point responses are attached below for your perusal. We trust that you find our revised manuscript better suited for publication in Vaccines than our original submission, and we look forward to your favorable response.

Yours sincerely,

HE Fang

Principal Investigator

Institute of Preventive Veterinary Medicine

College of Animal Sciences

Zhejiang University

866 Yuhangtang Road, Hangzhou

China

Reviewer #2 (minor concerns):

“Line 91. Introduce 'min'”

Page 3, Line 94---------Correction has been made accordingly.

“Line 99. Introduce 'h'”

Page 3, Line 102---------Correction has been made accordingly.

“Line 118. Substitute 2000 with '2,000'”

Page 3, Line 124---------Correction has been made accordingly.

“Line 138. Introduce 'hpi'”

Page 4, Line 143---------Correction has been made accordingly.

“Line 149. Remove the space within Δ”

Page 4, Line 155---------Correction has been made accordingly.

“Line 170. Please add also in the figure legend for how many hours did you pre-incubate CD163 SRCR5-9 with PRRSV”

Page 5, Line 177-178---------Correction has been made accordingly.

“Figure 6. In which cell lines did you perform IFA?”

Page 10, Line 282-283---------Correction has been made accordingly.

“Discussion. Although I find it well written and easy to ready, I think that it should be highlightened a bit more that this is only an in vitro study, and more study in vivo are need to assess the real applicability of CD163-antibody therapy against PRRSV”

Page 13, Line 410-414---------Application of the antibodies has been discussed accordingly.